# LaDe: The First Comprehensive Last-mile Express Dataset from Industry

## Abstract

Real-world last-mile express datasets are crucial for research in logistics, supply chain management, and spatio-temporal data mining. Despite a plethora of algorithms developed to date, no widely accepted, publicly available last-mile express dataset exists to support research in this field. In this paper, we introduce `LaDe`, the first publicly available last-mile express dataset with millions of packages from the industry. `LaDe` has three unique characteristics: (1) *Large-scale*. It involves 10,677k packages of 21k couriers over 6 months of real-world operation. (2) *Comprehensive information*. It offers original package information, such as its location and time requirements, as well as task-event information, which records when and where the courier is while events such as task-accept and task-finish events happen. (3) *Diversity*. The dataset includes data from various scenarios, including package pick-up and delivery, and from multiple cities, each with its unique spatio-temporal patterns due to their distinct characteristics such as populations. We verify `LaDe` on three tasks by running several classical baseline models per task. We believe that the large-scale, comprehensive, diverse feature of `LaDe` can offer unparalleled opportunities to researchers in the supply chain community, data mining community, and beyond. The dataset homepage is publicly available at https://anonymous.4open.science/r/Anonymous-64B3/.

## 1 Introduction

Driven by increasing urbanization and e-commerce development, last-mile delivery has emerged as a critical research area with growing interest from scholars and practitioners. **Last-Mile Delivery**, as illustrated in Figure 1, is the package transport process that connects the depot and the customers, including both the package pick-up (Macioszek, 2018; Ranathunga et al., 2021) and delivery (Boysen et al., 2021; Ratnagiri et al., 2022) process. In addition to being a key to customer satisfaction, last-mile delivery is both the most expensive and time-consuming part of the shipping process (Olsson et al., 2019; Mangiaracina et al., 2019). Consequently, researchers from different fields, from logistics operation management to spatio-temporal data mining, have been consistently shedding light on problems in last-mile delivery in recent years. These problems include route planning (Zeng et al., 2019; Li et al., 2021; Almasan et al., 2022), Estimated Time of Arrival (ETA) prediction (Wu & Wu, 2019; de Araujo & Etemad, 2021; Gao et al., 2021), and route prediction (Zhang et al., 2019; Wen et al., 2021; 2022), etc. A quick search for "last-mile delivery" on Google Scholar returns over 19,400 papers since 2018.

Recent endeavors (Wu & Wu, 2019; de Araujo & Etemad, 2021; Gao et al., 2021) focus on leveraging machine/deep learning techniques for problems in last-mile delivery research. A critical prerequisite for those researches is the availability of high-quality, large-scale datasets. Since such datasets have the potential to significantly accelerate advancements in specific fields, such as ImageNet (Deng et al., 2009) for computer vision and GLUE (Wang et al., 2018) for natural language processing. Nonetheless, in the domain of last-mile background research, a multitude of algorithms have been devised, but there is still an absence of a widely recognized, publicly accessible dataset. Consequently, research in this field has become concentrated within a limited number of industrial research laboratories, thereby restricting transparency and hindering research progress. Moreover, the lack of public datasets also poses a hurdle for industry practitioners to develop advanced algorithms for last-mile delivery.

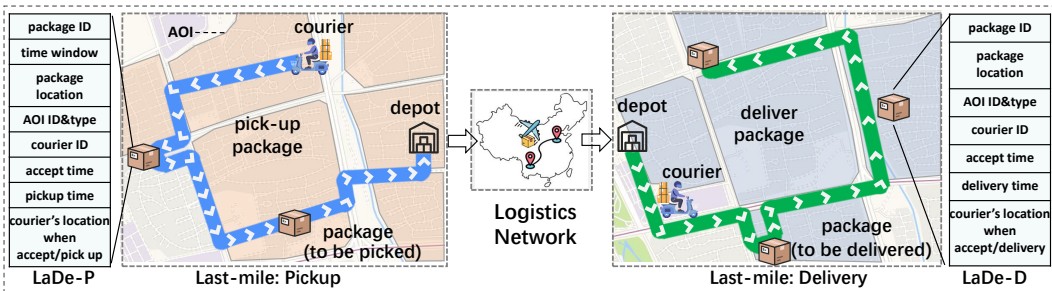

Figure 1: Overview of `LaDe` from last-mile express (better viewed in color), which includes two sub-datasets: `LaDe-P` from package pick-up process (i.e., couriers pick up packages from senders and return the depot) and `LaDe-D` from delivery process (i.e., couriers deliver packages from the depot to receivers).

To meet the rising calling for a public dataset, we propose `LaDe`, the first comprehensive Last-mile Express dataset collected by (company name blinded). It contains both package pick-up and delivery data as depicted in Figure 1. `LaDe` has several merits: (1) *Large-scale*, covering 10,677k packages of 21k couriers across 6 months. To the best of our knowledge, this is the largest publicly available dataset. (2) *Comprehensive*, providing detailed information on package, location, task-event, and courier. (3) *Diverse*, collecting data from both pick-up and delivery processes across various cities. By virtue of these advantages, `LaDe` can be employed to evaluate a wide spectrum of last-mile-related tasks. In this paper, we investigate its properties by three tasks, including route prediction (Zhang et al., 2019; Wen et al., 2021; 2022), estimated time of arrival prediction (Wu & Wu, 2019; de Araujo & Etemad, 2021; Gao et al., 2021), and spatio-temporal graph forecasting (Li et al., 2018; Yao et al., 2018; Bai et al., 2020). Beyond these tasks, it is easy to integrate some of the aforementioned features to support additional tasks. We believe that such a large-scale dataset like `LaDe` is a critical resource for developing advanced algorithms under the context of last-mile delivery, as well as for providing critical training and benchmarking data for learning-based algorithms. Overall, we identify three key contributions of this work:

- **A New Dataset.** We collect, process, and release `LaDe`. The dataset boasts large-scale, comprehensive, and diverse characteristics. To the best of our knowledge, it is the first exhaustive, industry-scale last-mile express dataset. The dataset is publicly accessible at https://anonymous.4open.science/r/Anonymous-64B3/.

- **Comprehensive Data Analysis.** Extensive data analysis is conducted to depict and highlight the properties of the dataset. Based on the analysis, we introduce potential tasks supported by `LaDe`, from logistics operation management to spatio-temporal data mining, and beyond.

- **Benchmark on Real-World Tasks.** We benchmark this dataset by performing three representative tasks, including service route prediction, estimated time of arrival prediction, and spatio-temporal graph forecasting. The source codes for these tasks are provided to promote research in this field.

The remainder of this paper is structured as follows. Section 2 discusses related work, and Section 3 introduces the details of the dataset, including the methodology used to construct the dataset, and the statistics and properties of the dataset. In Section 4, we benchmark the dataset on three tasks and discuss the potential use of the data in related research fields.

## 2 RELATED WORK

**Dataset Perspective.** To the best of our knowledge, there is no publicly available last-mile dataset containing both package pick-up and delivery data. The most relative effort comes from Amazon (Merchán et al., 2022) (named AmazonData in this paper). It is a courier-operated sequence dataset proposed for a last-mile routing research challenge hosted by Amazon. Specifically, this dataset contains 9,184 historical routes performed by Amazon couriers in 2018 in five metropolitan areas in the United States. Despite the contribution of AmazonData to the research field, it still has three

limitations: 1) Without pick-up data, it only contains data generated in the package delivery process; 2) Small scale, in terms of spatio-temporal range and the number of trajectories; 3) Lack of courier-related and task-event-related information, which prevents it from benefiting a wider group of researchers with different interests. In light of the above issues, we introduce an industry-scale, comprehensive dataset (i.e., `LaDe`) for researchers to develop and evaluate new ideas on real-world instances in last-mile delivery. The scale of `LaDe` is 5 times of AamazonData in terms of package number and 50 times in terms of trajectory number. We provide a detailed comparison of AamazonData and `LaDe` in Table 1.

Table 1: Comparison between `LaDe` and the related dataset.

| Dataset | Time span | #Trajectories | #Couriers | #Packages | Delivery Data | Pick-up Data | Courier Info | Task-event Info |
|---|---|---|---|---|---|---|---|---|
| AmazonData | 4 months | 9k | - | 2,182k | ✓ | × | × | × |
| LaDe | 6 months | 619k | 21k | 10,677k | ✓ | ✓ | ✓ | ✓ |

**Application Perspective.** Overall, last-mile logistics is an emerging interdisciplinary research area connecting transportation and AI technology, in which deep learning methods have long been the most popular model (Olsson et al., 2019). Broadly speaking, there are four branches in this field: 1) Emerging trends and technologies, which focus on technological solutions and innovations in last-mile logistics, such as courier's route and arrival time prediction (Wen et al., 2022; Gao et al., 2021), self-service technologies (Vakulenko et al., 2018), drone-assisted delivery (Taniguchi et al., 2020). 2) Last-mile-related data mining (Ruan et al., 2022b; 2020a), which aims to excavate the underlying patterns of knowledge from data generated by real-world operations for better logistics management. 3) Operational optimization, which focuses on optimizing last-mile operations and making better operational decisions, such as vehicle routing problem (Zeng et al., 2019; Breunig et al., 2019), delivery scheduling (Han et al., 2017), and facility location selection (Jahangiriesmaili et al., 2017; Kedia et al., 2020). 4) Supply chain structures, which focused on designing structures for last mile logistics, such as the network design (Lim & Srai, 2018). We refer readers to the paper (Olsson et al., 2019) for a more detailed, systematic classification of last-mile-related research. The proposed `LaDe` contains instances based on real operational data that researchers can use to advance the state-of-the-art in their fields and to expand its applications to industry settings.

## 3 PROPOSED DATASET: LADE

In this section, we formally introduce the `LaDe` Dataset. First, we describe the data collection process, followed by a detailed discussion of `LaDe`'s data fields and dataset statistics. Finally, we conduct a comprehensive analysis to highlight its unique properties. The dataset can be freely downloaded at https://anonymous.4open.science/r/Anonymous-64B3/ and noncommercially used with a custom license CC BY-NC 4.0[1].

### 3.1 DATA COLLECTION

This dataset is collected by (company name blinded), one of China's largest logistics platforms, which handles a tremendous volume of packages each day. A typical process for shipping a package involves the following steps: 1) The customer (sender) places a package pick-up order through the online platform. 2) The platform dispatches the order to an appropriate courier. 3) The courier picks up the package within the specified time window and returns to the depot (this constitutes the package pick-up process). 4) The package departs from the depot and traverses the logistics network until it reaches the target depot. 5) At the target depot, the delivery courier retrieves the package and delivers it to the recipient customer (known as the package delivery process). Among these steps, step 3 and 5 are referred to as the last-mile delivery, where couriers pick up/deliver packages from/to customers. Note that there is a notable difference between the pick-up and delivery scenarios. In the package delivery process, packages assigned to a particular courier are determined prior to the courier's departure from the depot. Conversely, in the pick-up process, packages assigned to a courier are not settled at the beginning. Rather, they are revealed over time, as customers can request pick-ups at any time. The dynamic nature of package pick-up presents substantial challenges in the

---

[1]https://creativecommons.org/licenses/by-nc/4.0/

research field. To advocate more efforts for the challenge and make the data more diverse, `LaDe` contains two sub-datasets in both pick-up and delivery scenarios, named `LaDe-P` and `LaDe-D`, respectively.

Specifically, we collect millions of package pick-up/delivery data generated in 6 months from different cities in China. To increase the diversity, we carefully selected 5 cities - Shanghai, Hangzhou, Chongqing, Jilin, and Yantai - which possess distinct characteristics such as populations, more details can be found in Table 9 of Appendix 7.2. A city contains different regions, with each region composed of several AOIs (Area of Interest) for logistics management. And a courier is responsible for picking up / delivering packages in several assigned AOIs. We give a simple illustration of the region-level and AOI-level segmentation of a city in Figure 2. To collect the data for each city, we first randomly select 30 regions in the city. Subsequently, we randomly sample couriers in each region and pick out all the selected couriers' picked-up/delivery packages during the 6 months. Note that when a courier is chosen, all his packages get selected. At the same time, all his packages fall within the randomly selected regions. Because a courier is responsible for several AOIs, they all belong to one of the selected regions.

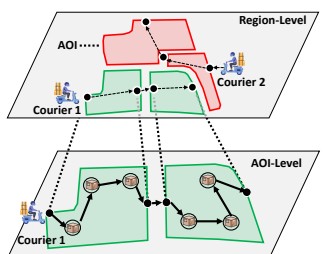

Figure 2: Region-level and AOI-level data.

**Privacy.** For the privacy issue, inspired by operations in geo-related data release work Merchán et al. (2022); Joshi et al. (2022), no customer-related information (such as the address, name, and ID) is contained in the dataset. And for the couriers, they are our staff, we have got their permission to collect, and analyze the dataset. Specifically, the following operations are adopted to further protect privacy: 1) a package is not linked to a customer in the dataset to protect the privacy of customers. 2) Instead of using the latitude and longitude, we utilize the coordinate of a package in a two-dimensional space $(x, y)$ to represent a package's location. The relative distance of two locations in real geographical space is preserved in the two-dimensional space. In this way, $(x, y)$ cannot be projected back to a real-world location, thus largely migrating the privacy issue. 3) For couriers, no sensitive information (e.g., gender and age) is included either.

## 3.2 DATASET DETAILS & STATISTICS

In this subsection, we present the dataset details and its basic statistics. A brief information on the data field is illustrated in Figure 1. And the detailed data field description of `LaDe-P` and `LaDe-D` can be found in Table 7 and Table 8 in Appendix 7.1 due to the page limitation.

To facilitate the utilization and analysis of the dataset, we transform and arrange each sub-dataset into tabular data presented in CSV format. Each record in this format contains relevant information pertaining to a picked-up or delivered package, primarily addressing the "who, where, when" aspects. Specifically, the record specifies which courier picked up or delivered the package, the location of the package, and the corresponding time. The recorded information can be broadly categorized into four types: 1) package information, which records the package ID and time windows requirements (if applicable); 2) stop information, recording the package's location information such as coordinates, AOI ID, and AOI type; 3) courier information, recording the courier's ID, and each courier is equipped with a personal digital assistant (PDA), which will consistently report the status of a courier (e.g., GPS) to the platform; 4) task-event information, recording the features of package accept, pick-up or delivery event, including when the event happens and the courier's location.

Overall, the package and task-event information can be recorded once the courier accepts the order, or finishes the order. Information about the stop comes from the geo-decoding system used in (company name blinded), which can parse the input location address into its corresponding coordinates with a given accuracy. Table 2 shows the statistics of the `LaDe-P`. Due to the page limitation, please refer to Table 10 in Appendix 7.2 for the statistics of the `LaDe-D`. Moreover, to intuitively illustrate the spatio-temporal characteristics of the dataset, we draw the spatial and temporal distribution of one city (Shanghai) in Figure 3 for one sub-dataset `LaDe-P`. From the Figure, we have the following observations. **Obs1:** Figure 3(a) shows that couriers' work time starts from 8:00 and ends at 19:00. The volume of package pick-up has a peak at 9:00 am and 5:00 pm, respectively. **Obs2:** Figure 3(b) and Figure 3(c) shows the spatial distribution of packages, where the distance between consecutive packages in a courier's route is usually within 1km. **Obs3:** Figure 3(d) shows the dis-

Table 2: Statistics of `LaDe-P`. AvgETA stands for the average arrival time per package. AvgPackage means the average package number of a courier per day. The unit of AvgETA is minute.

| City | Time span | Spatial span | #Trajectories | #Couriers | #Packages | #Location points | AvgETA | AvgPackage |
|---|---|---|---|---|---|---|---|---|
| Shanghai | 6 months | 20km×20km | 96k | 4,502 | 1,450k | 1,785k | 151 | 15.0 |
| Hangzhou | 6 months | 20km×20km | 119k | 5,347 | 2,130k | 2,427k | 146 | 17.8 |
| Chongqing | 6 months | 20km×20km | 83k | 2,982 | 1,172k | 1,475k | 140 | 14.0 |
| Yantai | 6 months | 20km×20km | 71k | 2,593 | 1,146k | 1,641k | 137 | 16.0 |
| Jilin | 6 months | 20km×20km | 18k | 665 | 261k | 399k | 123 | 13.8 |

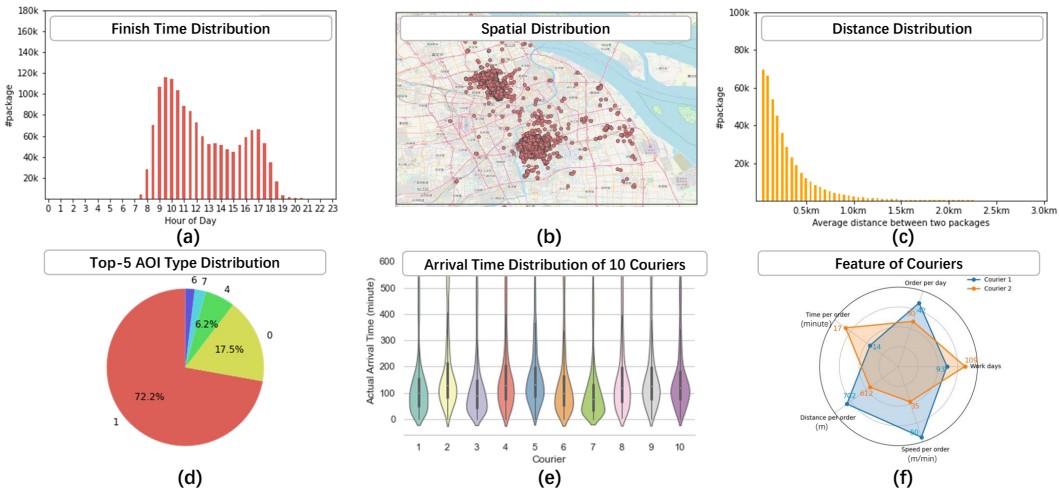

Figure 3: Spatial and temporal distribution of data in Shanghai of `LaDe-P`.

tribution of the top 5 AOI types in the data, illustrating that over 70% packages come from type 1. **Obs4:** Figure 3(e) shows the actual arrival time of 10 randomly selected couriers, from which we observed differences in the work efficiency of different couriers. It also shows that a majority of packages are picked up within 3 hours. **Obs5:** Figure 3(f) depicts the profile of two couriers in the dataset, where different characteristics such as work days, and average orders per day are observed.

### 3.3 DATASET PROPERTIES & CHALLENGES

In this subsection, we present our primary data analysis to highlight its properties and the challenges they entail.

**Large scale**. `LaDe` contains in total 10,667k packages and 619k trajectories that consist of 16,755k locations generated by 21k couriers, covering 5 cities over a total span of 6 months. The maximal package number of a courier one trip in the pick-up scenario and delivery scenario reaches 95 and 121, respectively. *Such large scale brings a significant challenge to algorithms in last-mile delivery.* To the best of our knowledge, this is the largest clean delivery dataset available to the research community, in terms of spatio-temporal coverage, the total number of packages, and the number of couriers' trajectories.

**Comprehensity**. `LaDe` aims to offer a wealth of information pertaining to last-mile delivery, encompassing various types of data such as detailed package information, task-event logs, courier trajectory details, and contextual features. The objective is to facilitate a broader range of research endeavors. *How to effectively leverage these comprehensive features to improve existing or inspire new tasks remains an open problem for researchers from different communities.*

**Diversity**. We increase the data's diversity from two perspectives: (1) scenario diversity – we introduce scenario diversity by collecting two sub-datasets representing both pick-up and delivery scenarios; (2) city diversity – we collect data from different cities to increase the diversity of the dataset. The cities in the dataset have different characteristics, leading to various spatio-temporal patterns in the dataset, where we give an illustration in Figure 4. For more information about the selected cities,

please refer to Table 9 in Appendix 7.1. *Such diversity brings the challenge of designing advanced models that can generalize well under cities with different characteristics.*

Figure 4: Diversity of cities. We select two cities, Hangzhou and Jilin, as an example to reveal their different spatio-temporal distributions. (a) The time distribution of packages in a day; (b) The ETA distribution of packages; (c) The distribution of the average distance between two consecutive packages in a courier's route. A significant difference is observed in the above illustration.

**Dynamism** (only for `LaDe-P`). Compared to `LaDe-D`, the tasks of a courier in `LaDe-P` are not settled at the beginning of the day. Rather, they are revealed along with the pick-up process as customers can place an order at any time. *Such dynamism in courier tasks poses significant technical challenges in various research areas*, with one notable example being dynamic route optimization Yao et al. (2019); Li et al. (2021).

Eqquiped with the above unique properties, `LaDe` offers the most extensive compilation of data for various research purposes background by last-mile delivery. It encompasses a variety of information across multiple domains, such as package details, event-based information, and courier information. Our aspiration is to make this abundant resource accessible to a broad spectrum of researchers, enabling them to undertake diverse and innovative studies.

## 4 APPLICATIONS

To prove `LaDe`'s ability to support multiple tasks, we benchmark the dataset in three learning-based tasks, including route prediction, estimated time of arrival prediction, and spatio-temporal graph forecasting. Those tasks all come from the real-world application and we illustrate them in Figure 5. The code is released at https://anonymous.4open.science/r/Anonymous-64B3/. Note that the dataset can support far more than the three tasks, which we envision more possible applications from different research fields at the end of the section. All methods were implemented with PyTorch 1.10.1 and Python 3.6.13, and deep learning methods were trained with an A40 GPU. The platform utilized is Ubuntu 23.04.

### 4.1 ROUTE PREDICTION

A crucial task in last-mile delivery services (such as logistics) is service route prediction Gao et al. (2021); Wen et al. (2022), which aims to estimate the future service route of a worker given his unfinished tasks at the request time.

**Problem Definition.** Formally, at a certain time $t$, a worker (i.e., courier) $w$ can have $n$ unfinished tasks, denoted by $\mathbf{X}_t^w = \{\mathbf{x}_1, \mathbf{x}_2, \ldots, \mathbf{x}_n\}$, where $\mathbf{x}_i$ corresponds to the feature vector of a task $i$. Given a worker $w$'s unfinished tasks at time $t$ and route constraints $\mathcal{C}$ (such as pick-up then delivery constraints), route prediction aims to learn a mapping function $\mathcal{F}_\mathcal{C}$ to predict the worker's future service route $\hat{\boldsymbol{\pi}}$ which can satisfy the given route constraints $\mathcal{C}$, formulated as: $\mathcal{F}_\mathcal{C}(\mathbf{X}_t^w) = [\pi_1, \pi_2 \cdots \pi_n]$, where $\pi_i$ means that the $i$-th node in the route is task $\pi_i$. And $\pi_i \in \{1, \cdots n\}$ and $\pi_i \neq \pi_j$ if $i \neq j$.

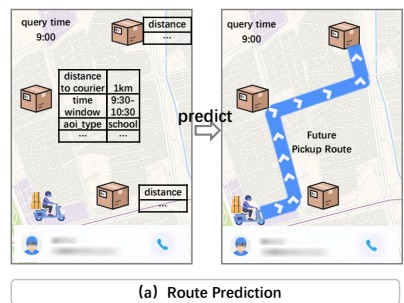 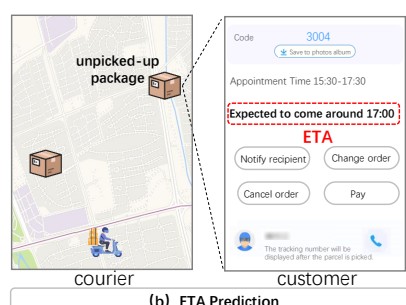 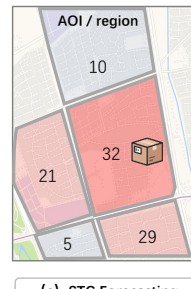

| (a) Route Prediction | (b) ETA Prediction | (c) STG Forecasting |

Figure 5: Illustration of three real-world applications. (a): Route prediction predicts the future pick-up route of a courier. (b): ETA prediction estimates the courier's arrival time for picking up or delivering packages. (c): STG forecasting predicts the future package number in given regions/AOIs.

**Dataset.** We choose `LaDe-P` as the dataset to conduct the experiment. The training, validation, and test set is split chronologically using a ratio of 6:2:2. Due to the space limit, we select three out of the five cities for conducting experiments, including Shanghai, Chongqing, and Yantai.

**Baselines & Hyperparameters.** We run six baselines on `LaDe`. 1) Basic methods: TimeGreedy Zhang et al. (2019) and DistanceGreedy Zhang et al. (2019). 2) Machine learning method: Osqure Zhang et al. (2019). 3) Deep learning models: DeepRoute Wen et al. (2021), FDNET Gao et al. (2021), Graph2Route Wen et al. (2022), and DRL4Route Mao et al. (2023). Hyperparameters search is performed on the validation set by evaluating hidden size in $\{16, 32, 64, 128\}$. We set the learning rate to 0.0001 and batch size to 64 for all deep-learning models. More details about the baselines and metrics can be found in Appendix 8.1.

**Results**. Following Wen et al. (2022), we adopt HR@$k$, KRC, LMD, and ED to evaluate model performance. Higher KRC, HR@$k$, and lower LSD and ED mean better performance. The number of packages in each sample is in $(0, 25]$. Table 3 shows the results of different methods on `LaDe`. It can be observed that basic models perform poorly since they can only make use of distance or time information. Deep models generally achieve better performance than shallow models, because of their ability to model abundant spatial and temporal features. This further proves the importance of the comprehensive information provided by `LaDe` for building more powerful models. Among deep models, Graph2Route performs well due to its ability to model the underlying graph correlation of different packages, while DRL4Route performs best since it utilizes deep reinforcement learning to solve the mismatch between the training and test criteria. More detailed and in-depth analysis can be found in Appendix 8.

Table 3: Experimental results of route prediction. We use bold and underlined fonts to denote the best and runner-up model, respectively.

| Method | Chongqing | | | | Shanghai | | | | Yantai | | | |
|---|---|---|---|---|---|---|---|---|---|---|---|---|
| | HR@3 ↑ | KRC ↑ | LSD ↓ | ED ↓ | HR@3 ↑ | KRC ↑ | LSD ↓ | ED ↓ | HR@3 ↑ | KRC ↑ | LSD ↓ | ED ↓ |
| TimeGreedy | 63.86 | 44.16 | 3.91 | 1.74 | 59.81 | 39.93 | 5.20 | 2.24 | 61.23 | 39.64 | 4.62 | 1.85 |
| DistanceGreedy | 62.99 | 41.48 | 4.22 | 1.60 | 61.07 | 42.84 | 5.35 | 1.94 | 62.34 | 40.82 | 4.49 | 1.64 |
| OR-Tools | 64.19 | 43.09 | 3.67 | 1.55 | 62.50 | 44.81 | 4.69 | 1.88 | 63.27 | 42.31 | 3.94 | 1.59 |
| Osqure | 71.55 | 54.53 | $\underline{2.63}$ | 1.54 | 70.63 | 54.48 | 3.27 | 1.92 | 70.41 | 52.90 | 2.87 | 1.59 |
| FDNET | 69.98 ± 0.32 | 52.07 ± 0.38 | 3.36 ± 0.04 | 1.51 ± 0.02 | 69.05 ± 1.41 | 52.72 ± 1.98 | 4.08 ± 0.29 | 1.86 ± 0.03 | 69.08 ± 0.61 | 50.62 ± 1.20 | 3.60 ± 0.15 | 1.57 ± 0.02 |
| DeepRoute | 72.09 ± 0.39 | 55.72 ± 0.40 | 2.66 ± 0.06 | 1.51 ± 0.01 | 71.66 ± 0.11 | 56.20 ± 0.27 | 3.26 ± 0.08 | 1.86 ± 0.01 | 71.44 ± 0.28 | 54.74 ± 0.49 | 2.80 ± 0.02 | $\underline{1.53 ± 0.02}$ |
| Graph2Route | 72.31 ± 0.20 | 56.08 ± 0.14 | $\underline{2.53 ± 0.01}$ | $\underline{1.50 ± 0.01}$ | 71.69 ± 0.12 | 56.53 ± 0.12 | 3.12 ± 0.01 | 1.86 ± 0.01 | 71.52 ± 0.14 | 55.02 ± 0.10 | $\underline{2.71 ± 0.01}$ | 1.54 ± 0.01 |
| DRL4Route | **73.12 ± 0.09** | **57.23 ± 0.17** | **2.43 ± 0.02** | **1.48 ± 0.01** | **72.18 ± 0.18** | **57.20 ± 0.20** | **3.06 ± 0.02** | **1.84 ± 0.01** | **72.21 ± 0.08** | **55.94 ± 0.13** | **2.62 ± 0.01** | **1.51 ± 0.01** |

## 4.2 ESTIMATED TIME OF ARRIVAL PREDICTION

Estimated Time of Arrival (ETA) prediction aims to forecast when the task is going to be finished, e.g., the delivery time of a package. It is one of the most important tasks in many delivery platforms since it directly influences customers' experience Wu & Wu (2019).

**Problem Definition.** Given an ETA query of worker $w$ at time $t$, i.e., $q = \{t, \mathbf{X}_t^w\}$, where $\mathbf{X}_t^w = \{\mathbf{x}_1, \mathbf{x}_2, \ldots, \mathbf{x}_n\}$ is the courier's unfinished packages, ETA prediction aims to build a model $\mathcal{F}$ that can map the input query to the arrival time (i.e., pick-up/delivery time) $Y$ for the unfinished package set: $\mathcal{F}(q) \mapsto Y = \{y_1, \ldots, y_n\}$, where $y_i = t_i^{\text{actual}} - t$ and $t^{\text{actual}}$ is task $i$'s actual arrival time.

**Dataset.** `LaDe-D` is utilized for this experiment (note that `LaDe-P` can also be used for this task). We split the data into training, validation, and test sets chronologically in a ratio of 6:2:2.

**Baselines & Hyperparameters.** Six baselines are evaluated for the task, including a simple speed-based method SPEED, machine learning methods LightGBM Ke et al. (2017) and KNN Song et al. (2019), and deep models Multi-Layer Perceptron (MLP), FDNET Gao et al. (2021), and RAN-KETPA Wen et al. (2023). We also perform hyperparameters search on the validation set by hidden size in $\{16, 32, 64, 128\}$ for all deep models. The learning rate and batch size are set to 0.00005 and 32 for all models. See more details in Appendix 8.2.

**Results.** MAE, RMSE, and ACC@20 are used to evaluate the performance of time prediction models. Higher ACC@20 and lower MAE and RMSE indicate better performance. From the results shown in Table 4, we can see that learning-based models outperform SPEED by a large margin because of their ability to model multiple spatio-temporal factors. We also observe a huge performance gap of the same method in different cities. For example, the best model, RANKETPA, achieves 70% in terms of ACC@20 in Shanghai, while it gets a much lower accuracy of 51% in the other two datasets. It deserves further study to build a more powerful model that can generalize well in cities with different properties.

Table 4: Experiment results of ETA prediction.

| Method | Shanghai | | | Chongqing | | | Yantai | | |
|---|---|---|---|---|---|---|---|---|---|
| | MAE ↓ | RMSE ↓ | ACC@20 ↑ | MAE ↓ | RMSE ↓ | ACC@20 ↑ | MAE ↓ | RMSE ↓ | ACC@20 ↑ |
| LightGBM | 17.48 | 20.39 | 0.68 | 24.78 | 28.64 | 0.47 | **23.16** | **27.29** | **0.52** |
| SPEED | 23.75 | 27.86 | 0.58 | 33.42 | 39.45 | 0.45 | 31.41 | 37.09 | 0.46 |
| KNN | 21.28 | 25.36 | 0.60 | 30.05 | 35.49 | 0.42 | 28.96 | 34.48 | 0.44 |
| MLP | $18.58_{\pm 0.37}$ | $21.54_{\pm 0.34}$ | $0.66_{\pm 0.02}$ | $29.75_{\pm 0.92}$ | $34.62_{\pm 1.25}$ | $0.51_{\pm 0.01}$ | $25.84_{\pm 0.23}$ | $29.67_{\pm 0.19}$ | $0.45_{\pm 0.01}$ |
| FDNET | $18.47_{\pm 0.31}$ | $21.44_{\pm 0.34}$ | $0.67_{\pm 0.02}$ | $28.17_{\pm 1.30}$ | $32.60_{\pm 1.52}$ | $0.46_{\pm 0.02}$ | $27.08_{\pm 3.24}$ | $31.15_{\pm 4.01}$ | $0.45_{\pm 0.01}$ |
| RANKETPA | $\mathbf{17.18_{\pm 0.06}}$ | $\mathbf{20.18_{\pm 0.08}}$ | $\mathbf{0.70_{\pm 0.01}}$ | $\mathbf{24.00_{\pm 0.31}}$ | $\mathbf{27.89_{\pm 0.33}}$ | $\mathbf{0.51_{\pm 0.01}}$ | $23.62_{\pm 0.03}$ | $27.52_{\pm 0.04}$ | $0.51_{\pm 0.01}$ |

## 4.3 SPATIO-TEMPORAL GRAPH (STG) FORECASTING

`LaDe` contains the package data with information that records when and where the package order is placed. Based on this, the package number of a region within a certain period can be calculated. In this way, `LaDe` also contributes as a new dataset to another well-known task – *spatio-temporal graph forecasting* Li et al. (2018); Yao et al. (2018); Simeunović et al. (2021), which aims to predict future graph signals given its historical observations.

**Problem Definition.** Let $\mathcal{G} = \{\mathcal{V}, \mathcal{E}, \mathbf{A}\}$ represent a graph with $V$ nodes, where $\mathcal{V}, \mathcal{E}$ are the node set and edge set, respectively. $\mathbf{A} \in \mathbb{R}^{V \times V}$ is a weighted adjacency matrix to describe the graph topology. For $\mathcal{V} = \{v_1, \ldots, v_V\}$, let $\mathbf{x}_t \in \mathbb{R}^{F \times V}$ denote $F$-dimensional signals generated by the $V$ nodes at time $t$. Given historical graph signals $\boldsymbol{x}^{\mathrm{h}} = [\mathbf{x}_1, \cdots, \mathbf{x}_{T_h}]$ of $T_h$ time steps and the graph $\mathcal{G}$ as inputs, STG forcasting aims at learning a function $\mathcal{F}$ to predict future graph signals $\boldsymbol{x}^{\mathrm{p}}$, formulated as: $\mathcal{F} : (\boldsymbol{x}^{\mathrm{h}}; \mathcal{G}) \rightarrow [\mathbf{x}_{T_h+1}, \cdots, \mathbf{x}_{T_h+T_p}] := \boldsymbol{x}^{\mathrm{p}}$, where $T_p$ is the forecasting horizon.

**Dataset.** `LaDe-P` is used to conduct this experiment. More experiment details can be found in Appendix 8.3. Each node corresponds to a region within the city. The signal of each node represents the number of packages picked up during a particular time stamp. We set the time interval to be 1 hour. Our objective is to leverage the data from the previous 24 hours to predict the package volume for the subsequent 24 hours. We use the ratio of 6:2:2 for training, evaluation, and testing sets based on the chronological order of the timestamps.

**Baselines & Hyperparameters.** We evaluate eight baselines, including a traditional method (i.e., HA Zhang et al. (2017)), and recent deep learning models, including DCRNN Li et al. (2018), STGCN Yu et al. (2018), GWNET Wu et al. (2019), ASTGCN Guo et al. (2019), MTGNN Wu et al. (2020), AGCRN Bai et al. (2020), STGNCDE Choi et al. (2022) and GMSDR Liu et al. (2022). We set the hidden size, learning rate, and batch size to 32, 0.001, and 32 for all models.

**Results.** MAE and RMSE are used as the metrics, and results are shown in Table 5. According to the results, the traditional HA model consistently shows suboptimal results across all regions, indicating its limitations in the STG forecasting tasks. In the Shanghai dataset, STGCN emerges as the most effective, emphasizing the utility of Temporal Convolutional Networks (TCNs) in this context. In Hangzhou, AGCRN displays commendable accuracy, surpassing its predecessor, DCRNN, underscoring the advancements in RNN-based predictions. In Chongqing, while the newer models

like STGNCDE, and GMSDR, introduced in 2022, are promising, they don't always outperform the established methodologies, which suggests that the optimal model choice is closely tied to the specific dynamics and characteristics of each region.

Table 5: Experimental results of spatio-temporal graph prediction.

| Method | Shanghai | | Hangzhou | | Chongqing | |
|---|---|---|---|---|---|---|
| | MAE ↓ | RMSE ↓ | MAE ↓ | RMSE ↓ | MAE ↓ | RMSE ↓ |
| HA Zhang et al. (2017) | 4.63 | 9.91 | 4.78 | 10.53 | 2.44 | 5.30 |
| DCRNN Li et al. (2018) | $3.69 \pm 0.09$ | $7.08 \pm 0.12$ | $4.14 \pm 0.02$ | $7.35 \pm 0.07$ | $2.75 \pm 0.07$ | $5.11 \pm 0.12$ |
| STGCN Yu et al. (2018) | $\mathbf{3.04} \pm \mathbf{0.02}$ | $\mathbf{6.42} \pm \mathbf{0.05}$ | $\underline{3.01} \pm 0.04$ | $\underline{5.98} \pm 0.10$ | $2.16 \pm 0.01$ | $4.38 \pm 0.03$ |
| GWNET Wu et al. (2019) | $3.16 \pm 0.06$ | $6.56 \pm 0.11$ | $3.22 \pm 0.03$ | $6.32 \pm 0.04$ | $2.22 \pm 0.03$ | $4.45 \pm 0.05$ |
| ASTGCN Guo et al. (2019) | $\underline{3.12} \pm 0.06$ | $\underline{6.48} \pm 0.14$ | $3.09 \pm 0.04$ | $6.06 \pm 0.10$ | $\mathbf{2.11} \pm \mathbf{0.02}$ | $\mathbf{4.24} \pm \mathbf{0.03}$ |
| MTGNN Wu et al. (2020) | $3.13 \pm 0.04$ | $6.51 \pm 0.13$ | $\mathbf{3.01} \pm \mathbf{0.01}$ | $\mathbf{5.83} \pm \mathbf{0.03}$ | $\underline{2.15} \pm 0.01$ | $\underline{4.28} \pm 0.05$ |
| AGCRN Bai et al. (2020) | $3.93 \pm 0.03$ | $7.99 \pm 0.08$ | $4.00 \pm 0.03$ | $7.88 \pm 0.06$ | $2.46 \pm 0.00$ | $4.87 \pm 0.01$ |
| STGNCDE Choi et al. (2022) | $3.74 \pm 0.15$ | $7.27 \pm 0.16$ | $3.55 \pm 0.04$ | $6.88 \pm 0.10$ | $2.32 \pm 0.07$ | $4.52 \pm 0.07$ |
| GMSDR Liu et al. (2022) | $3.70 \pm 0.10$ | $7.16 \pm 0.91$ | $3.73 \pm 0.28$ | $7.18 \pm 0.38$ | $2.38 \pm 0.09$ | $4.88 \pm 0.21$ |

## 4.4 DISSCUSSION OF OTHER POTENTIAL TASKS

In addition to primary tasks, the dataset can provide substantial support for a wide range of other tasks in different research fields. Firstly, `LaDe` can be used for spatial-temporal data (STD) representation learning, which involves many topics that can be broadly classified by the representation object: i) POI (Point of Interest) representation learning Lin et al. (2021), i.e., to learn the representation of the pickup/delivery location. ii) trajectory representation learning, i.e., to learn the representation of courier trajectory Fu & Lee (2020); ii) AOI representation learning Yue et al. (2021). The models developed in LaDe can also be generalized to other fields such as food delivery and riding sharing. Secondly, `LaDe` can be utilized to verify algorithms for optimization problems, such as the vehicle route problem Zeng et al. (2019), delivery scheduling problem Han et al. (2017). Thirdly, it can be used to data mining tasks within the context of last-mile delivery Ji et al. (2019); Ruan et al. (2022b) and sptial crowding sourcing Han et al. (2017); Chen et al. (2020).

Moreover, benefited by its large data volume and detailed information, `LaDe` shows great potential to support the development of foundation models Bommasani et al. (2021) in geo-related domains Wu et al. (2023). In summary, we present a list of tasks supported by `LaDe` in Table 6, highlighting the minimal required information necessary for performing each task using `LaDe`. This effectively showcases `LaDe`'s remarkable multi-task support capability. In the future, we plan to explore a wider range of applications on `LaDe`.

Table 6: Supported tasks with the minimal required information.

| Task | Package Info | Stop Info | Courier Info | Task-event Info | Context |
|---|---|---|---|---|---|
| STD Representation Learning Lin et al. (2021) | ✓ | | | ✓ | |
| Vehicle Routing Zeng et al. (2019) | ✓ | ✓ | | ✓ | |
| Delivery Scheduling Han et al. (2017) | ✓ | ✓ | | ✓ | |
| Last-Mile Data Mining Ji et al. (2019); Ruan et al. (2022b) | ✓ | ✓ | ✓ | ✓ | ✓ |
| Spatial Crowdsourcing Han et al. (2017); Chen et al. (2020) | ✓ | ✓ | ✓ | ✓ | |
| Time Prediction Ruan et al. (2020b; 2022a) | ✓ | ✓ | ✓ | ✓ | |
| Route Prediction Gao et al. (2021); Wen et al. (2022) | ✓ | ✓ | ✓ | ✓ | |
| STG Forecasting Yao et al. (2018); Simeunović et al. (2021) | ✓ | ✓ | | | ✓ |

## 5 CONCLUSION

In this paper, we introduced `LaDe`, the first comprehensive industry-scale last-mile express dataset, addressing the lack of a widely accepted, publicly available dataset for last-mile delivery research. `LaDe` provides a critical resource for researchers and practitioners to develop advanced algorithms in the context of last-mile delivery, with its large-scale, comprehensive, diverse, and dynamic characteristics enabling it to serve as a new and challenging benchmark dataset. We have also demonstrated the versatility of `LaDe` by benchmarking it on three real-world tasks, showcasing its potential applications in various research fields. The source code is released along with the dataset to drive the development of this area. By releasing `LaDe`, we aim to promote further research and collaboration among researchers from different fields, encouraging them to utilize it for developing novel algorithms and models, as well as comparing and validating their methods against state-of-the-art approaches. We believe that `LaDe` will significantly contribute to ongoing efforts to improve efficiency, cost-effectiveness, and customer satisfaction in last-mile delivery, ultimately benefiting the research community and logistics industry.

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

# 6 APPENDIX

# 7 DETAILED DATASET DESCRIPTION

## 7.1 DATA FIELD

Table 7: Description of data fields of `LaDe-P`.

| Data field | Description | Unit/format |
|---|---|---|
| Package information | | |
| package_id | Unique identifier of each package | Id |
| time_window_start | start of the required time window | Time |
| time_window_end | end of the required time window | Time |
| Stop information | | |
| x/y | Coordinates of each stop in the two-dimensional space | Float |
| city | City | String |
| region_id | Id of the Region | String |
| aoi_id | Id of the AOI (Area of Interest) | Id |
| aoi_type | Type of the AOI | Categorical |
| Courier Information | | |
| courier_id | Id of the courier | Id |
| Task-event Information | | |
| accept_time | The time when the courier accepts the task | Time |
| accept_gps_time | The time of the GPS point whose time is the closest to accept time | Time |
| accept_gps_x/accept_gps_y | Coordinates when the courier accept the task | Float |
| pickup_time | The time when the courier picks up the task | Time |
| pickup_gps_time | The time of the GPS point whose time is the closest to the pickup_time | Time |
| pickup_gps_x/pickup_gps_y | Coordinates when the courier picks up the task | Float |
| Context information | | |
| ds | the date of the package pickup | Date |

Table 8: Description of data fields of `LaDe-D`.

| Data field | Description | Unit/format |
|---|---|---|
| Package information | | |
| package_id | Unique identifier of each package | Id |
| Stop information | | |
| x/y | Coordinates of each stop in the two-dimensional space | Float |
| city | City | String |
| region_id | Id of the region | Id |
| aoi_id | Id of the AOI | Id |
| aoi_type | Type of the AOI | Categorical |
| Courier Information | | |
| courier_id | Id of the courier | Id |
| Task-event Information | | |
| accept_time | The time when the courier accepts the task | Time |
| accept_gps_time | The time of the GPS point whose time is the closest to accept time | Time |
| accept_gps_x/accept_gps_y | Coordinates when the courier accept the task | Float |
| delivery_time | The time when courier finishes delivering the task | Time |
| delivery_gps_time | The time of the GPS point whose time is the closest to the got time | Time |
| delivery_gps_x/delivery_gps_y | Coordinates when the courier finishes the task | Float |
| Context information | | |
| ds | the date of the package delivery | Date |

Table 9: Information of different selected cities.

| City | Description |
|---|---|
| Shanghai | One of the most prosperous cities in China, with a large number of orders per day. |
| Hangzhou | A big city with well-developed online e-commerce and a large number of orders per day. |
| Chongqing | A big city with complicated road conditions in China, with a large number of orders. |
| Jilin | A middle-size city in China, with a small number of orders each day. |
| Yantai | A small city in China, with a small number of orders every day. |

## 7.2 DATA STATISTICS

Table 10 shows the detailed statistics of `LaDe-D`.

Table 10: Statistics of `LaDe-D`. AvgETA stands for the average arrival time per package. AvgPackage means the average package number of a courier per day. The unit of AvgETA is minute.

| City | Time span | Spatial span | #Trajectories | #Couriers | #Packages | #GPS points | AvgETA | AvgPackage |
|---|---|---|---|---|---|---|---|---|
| Shanghai | 6 months | 20km×20km | 70k | 1,733 | 1,483k | 2,967k | 102 | 21.1 |
| Hangzhou | 6 months | 20km×20km | 71k | 1,392 | 1,861k | 3,723k | 147 | 25.9 |
| Chongqing | 6 months | 20km×20km | 68k | 1,494 | 931k | 1,862k | 182 | 13.5 |
| Yantai | 6 months | 20km×20km | 17k | 205 | 206k | 410k | 244 | 11.5 |
| Jilin | 6 months | 20km×20km | 2k | 57 | 31k | 61k | 203 | 16.2 |

## 8 EXPERIMENTS DETAILS

### 8.1 EXPERIMENT DETAILS OF ROUTE PREDICTION

**Methods.** We adopt the following methods for experiments:

- TimeGreedy Zhang et al. (2019): A greedy algorithm, which ranks all the candidate tasks by sorting their remaining time.

- DistanceGreedy Zhang et al. (2019): A greedy algorithm, which chooses to take the nearest package at each step, regardless of time requirements and other factors.

- OR-Tools Bello et al. (2017) adopts a heuristic strategy to search the route with the minimum travel distance as worker's future route.

- Osqure Zhang et al. (2019): A machine learning method, which predicts the next package at each time step through a machine learning algorithm, by considering it as a multi-class classification problem.

- DeepRoute Wen et al. (2021): A deep learning method, equipped with a Transformer encoder and Pointer Net decoder.

- FDNET Gao et al. (2021): A deep learning method, equipped with a Bi-LSTM encoder and Pointer Net decoder.

- Graph2Route Wen et al. (2022): A deep learning method, equipped with a dynamic graph encoder and personalized route decoder.

- DRL4Route Mao et al. (2023): A deep reinforcement learning method that introduces the non-differentiable metric as the reward for training the route prediction agent.

**Metrics.** Following the setting in Wen et al. (2022), the following metrics are utilized to evaluate the performance of route prediction methods:

- **KRC**: Kendall Rank Correlation Kendall (1938) is a statistical metric to measure the ordinal association between two sequences. Let $\hat{Y}$ and $Y$ be two sequences and $R_{\hat{Y}}(i) \in [1, |Y|]$ be the position of item $i$ in $Y$, a node pair $(i, j)$ is said to be concordant if and only if both $R_{\hat{Y}}(i) > R_{\hat{Y}}(j)$ and $R_Y(i) > R_Y(j)$, or both $R_{\hat{Y}}(i) < R_{\hat{Y}}(j)$ and $R_Y(i) < R_Y(j)$. Otherwise, it is said to be discordant. To calculate this metric, nodes in the prediction are first divided into two

sets: i) nodes in label $\mathcal{V}_{in} = \{\hat{y}_i | \hat{y}_i \in Y\}$, and ii) nodes not in label $\mathcal{V}_{not} = \{\hat{y}_i | \hat{y}_i \notin Y\}$. The order of items in $\mathcal{V}_{in}$ is available, while it is hard to tell the order of items in $\mathcal{V}_{not}$. Still, we know that all items in $\mathcal{V}_{in}$ are ahead of that in $\mathcal{V}_{not}$. Therefore, we compare the nodes pairs $\{(i,j)|i,j \in \mathcal{V}_{in}$ and $i \neq j\} \cup \{(i,j)|i \in \mathcal{V}_{in}$ and $j \in \mathcal{V}_{not}\}$. To this end, KRC is defined as:

$$\text{KRC} = \frac{N_c - N_d}{N_c + N_d}, \tag{1}$$

where $N_c$ is the number of concordant pairs, and $N_d$ is the number of discordant pairs.

- **ED:** Edit Distance Nerbonne et al. (1999) (ED) is an indicator to quantify how dissimilar two sequences $Y$ and $\hat{Y}$ are to one another, by counting the minimum number of required operations to transform one sequence into another.

- **LSD**: Location Square Deviation (LSD) measures the degree that the prediction deviates from the label, formulated as:

$$\text{LMD} = \frac{1}{m} \sum_{i=1}^{m} |(R_Y(i) - R_{\hat{Y}}(i))|. \tag{2}$$

- **HR@$k$**: Hit-Rate@$k$ quantifies the similarity between the top-$k$ items of two sequences. It describes how many of the first $k$ predictions are in the label, which is formulated as follows:

$$\text{HR@}k = \frac{\hat{Y}_{[1:k]} \cap Y_{[1:k]}}{k}. \tag{3}$$

**Evaluation Results.** We conduct route prediction experiments in three cities in China to analyze the potential ability of LaDe to support intelligent last-mile express service, as well as provide the community with the cookbook for designing advanced algorithms for downstream tasks in logistics.

Specifically, heuristic algorithms such as TimeGreedy and DistanceGreedy perform relatively poorly since the worker's route is influenced by various spatio-temporal factors such as distance and end of the required time window. OR-Tools performs inferior in route prediction compared with machine learning methods. It proves that the worker's route is not always the shortest given a set of unfinished tasks. Graph2Route performs relatively well because it leverages the spatio-temporal correlations by graph neural networks and introduces graph structure information in the decoder.

However, deep learning methods such as FDNET, DeepRoute, and Graph2Route utilize cross entropy as training criteria. In the test phase, the non-differentiable objective, such as LSD, is designed to evaluate the predicted route. Thus, the training criteria are different from the test one, which could trim down their performance when applied in a real-world system. To address this problem and achieve more accurate route prediction, DRL4Route combines the power of reinforcement learning methods in non-differentiable objective optimization with the abilities of deep learning models in behavior learning.

## 8.2 EXPERIMENT DETAILS OF TIME PREDICTION

**Methods.** The following methods are chosen for experiments:

- SPEED, a simple speed-based method that utilizes distance/speed as the prediction value, where speed is calculated based on each worker's history trajectories. We set the speed for workers without previous trajectories as the average speed calculated by all workers.

- LightGBM Ke et al. (2017), a popular machine-learning method for regression tasks.

- KNN Song et al. (2019), a machine-learning method that trains a regressor based on K-Nearest Neighbors algorithm to predict the arrival time.

- MLP Popescu et al. (2009), a deep neural network model with 2 layers of MLPs.

- FDNET Gao et al. (2021), a deep model that predicts both route and time of unfinished tasks.

- RANKETPA Wen et al. (2023), a two-step model that first predicts the route, based on which the time prediction is conducted.

**Metrics.** MAE (Mean Absolute Error) and RMSE (Root Mean Squared Error), and ACC@20 are utilized as metrics. Note that delivery platforms usually provide an interval of arrival time for customer notification. Thus we compute the ratio of prediction where the time difference between predicted time and true time is less than 20 minutes (ACC@30), formulated as ACC@20 = $\frac{1}{N} \sum_{i=1}^{N} \mathbb{I}(|\hat{y_i} - y_i| < 30)$.

**Evaluation Results.** We conduct experiments using LaDe-D in three cities to analyze potential improvements for more accurate delivery time prediction. Table 4 summarizes the overall performance comparison. Specifically, SPEED is intuitive and efficient but inaccurate since complex factors influencing the delivery time, such as accept time and type of the AOI, are not considered. Even though the available features are more scarce in LaDe-D than in LaDe-P, learning-based methods outperform SPEED by a lot margin. It further proves that mining spatio-temporal correlations help better delivery time prediction.

It is also conducive to modeling the worker's route first and then predicting the delivery time, as suggested by the comparative results of RANKETPA. RANKETPA performs relatively better in the dataset collected from Shanghai and Chongqing. However, in the dataset collected from Yantai, LightGBM performs better than RANKETPA mainly because the number of trajectories is less abundant and diverse in Yantai compared to Shanghai and Chongqing, which could be hard for training a sophisticated deep learning model. From the results of the six methods, the error in time prediction is still huge. More advanced algorithms are waiting to be explored to support delivery time prediction.

### 8.3 EXPERIMENT DETAILS OF SPATIO-TEMPORAL GRAPH FORECASTING

**Methods.** For our Spatio-temporal Graph Forecasting experimental setup, we have selected the following methods:

- **HA** Zhang et al. (2017): HA predicts future values of a time series by calculating the mean of past observations that correspond to the same time periods.

- **DCRNN** Li et al. (2018): DCRNN employs a neural network architecture that incorporates diffusion convolution and sequence-to-sequence mechanisms. This enables the model to effectively learn spatial dependencies and temporal relations within the data.

- **STGCN** Yu et al. (2018): STGCN is a specialized spatio-temporal graph convolution network that synergistically merges spectral graph convolution with 1D convolution. This unique combination allows the model to effectively capture correlations between spatial and temporal dimensions, enabling a comprehensive understanding of the interplay between space and time in the data.

- **GWNET** Wu et al. (2019): GWNET creates an adaptive adjacency matrix to capture spatial correlations and uses 1D dilated causal convolution to capture temporal dependence.

- **ASTGCN** Guo et al. (2019): ASTGCN leverages the power of attention-based mechanisms and a spatio-temporal convolution system to dynamically capture spatio-temporal correlations within the data. By incorporating attention, the model can focus on relevant information and effectively model various temporal properties of traffic flows.

- **MTGNN** Wu et al. (2020): MTGNN adopts a message-passing framework to effectively model the temporal dynamics of graph-structured data. It achieves this by aggregating information from spatially neighboring nodes and past time steps. By leveraging this approach, MTGNN captures the interdependencies and changes over time, enabling a comprehensive understanding of the data's temporal dynamics.

- **AGCRN** Bai et al. (2020): AGCRN incorporates two key modules, namely Node Adaptive Parameter Learning and Data Adaptive Graph Generation, to automatically infer inter-dependencies in traffic series and capture node-specific patterns.

- **GMSDR** Liu et al. (2022): GMSDR improves upon RNNs by incorporating the hidden states of multiple historical time steps as input at each time unit.

- **STGNCDE** Choi et al. (2022): STGNCDE is an innovative spatio-temporal graph neural controlled differential equation model that leverages two neural control differential equations to process both spatial and sequential data.

**Metrics.** To assess the performance of the above-mentioned models in spatio-temporal graph forecasting on our dataset, we employ the metrics of Mean Absolute Error (MAE) and Root Mean Squared Error (RMSE).

# 9 DATASHEET OF DATASET

## 9.1 MOTIVATION

- **For what purpose was the dataset created?** Was there a specific task in mind? Was there a specific gap that needed to be filled? Please provide a description.

To meet the rising calling for datasets in the field of last-mile delivery research, we propose LaDe, the first industry-scale multipurpose real-world dataset. Compared with existing public datasets, LaDe has serval merits: (1) large-scale, it consists of millions of packages, which can serve as a data foundation for learning-based algorithms in last-mile delivery. (2) Comprehensive information, the dataset contains more comprehensive features, which enables the data to support multiple research tasks. (3) Scenario diversity, it contains the data from both the package pick-up and delivery scenarios. Researchers can use the two sub-datasets to study the different work patterns of couriers in different scenarios.

- **Who created the dataset (e.g., which team, research group) and on behalf of which entity (e.g., company, institution, organization)?**

The dataset was created by Artificial Intelligence Department, (company name blinded).

- **Who funded the creation of the dataset?** If there is an associated grant, please provide the name of the grantor and the grant name and number.

No.

## 9.2 COMPOSITION

- **What do the instances that comprise the dataset represent (e.g., documents, photos, people, countries)?** Are there multiple types of instances (e.g., movies, users, and ratings; people and interactions between them; nodes and edges)? Please provide a description.

The instances are packages picked up/delivered in the last-mile delivery.

- **How many instances are there in total (of each type, if appropriate)?**

There are 10,667k instances in LaDe, where an instance represents the features of a package.

- **Does the dataset contain all possible instances or is it a sample (not necessarily random) of instances from a larger set?** If the dataset is a sample, then what is the larger set? Is the sample representative of the larger set (e.g., geographic coverage)? If so, please describe how this representativeness was validated/verified. If it is not representative of the larger set, please describe why not (e.g., to cover a more diverse range of instances, because instances were withheld or unavailable).

The dataset is a sample of instances. We first randomly select serval regions in a city, then collect all the packages in that region within a certain period. Note that for each region, the dataset contains all possible instances within the given time period. To further increase the diversity of the dataset, five cities with different populations are selected and recorded.

- **What data does each instance consist of?** "Raw" data (e.g., unprocessed text or images) or features? In either case, please provide a description.

The format of each instance in LaDe-P is (*package_id, time_window_start, time_window_end, lng, lat, city, aoi_id, aoi_type, courier_id, accept_time, accept_gps_time, accept_gps_x, accept_gps_y, pickup_time, pickup_gps_time, pickup_gps_x, pickup_gps_y, ds*).

The format of each instance in `LaDe-D` is *(package_id, lng, lat, city, aoi_id, aoi_type, courier_id, accept_time, accept_gps_time, accept_gps_x, accept_gps_y, delivery_time, delivery_gps_time, delivery_gps_x, delivery_gps_y, ds)*.

For the detailed description of each field, please refer to Table 7 and Table 8 in Appendix 7.1.

- **Is there a label or target associated with each instance?** If so, please provide a description.

Since the dataset is proposed to support multiple tasks in last-mile delivery, for easy use and flexibility, a label for a specific task is not contained in one instance. However, it is easy to construct the label for different research purposes from the raw information. Take the estimated time of arrival prediction as an example. The actual arrival time (in this case, the label) can be calculated by the difference between the got_time and query_time.

- **Is any information missing from individual instances?** If so, please provide a description, explaining why this information is missing (e.g., because it was unavailable). This does not include intentionally removed information, but might include, e.g., redacted text.

Some instances lack the courier's location when accepting/finishing the package, i.e., accept_gps_x, accept_gps_y. The corresponding information is massing in the real system.

- **Are there recommended data splits (e.g., training, development/validation, testing)?** If so, please provide a description of these splits, explaining the rationale behind them.

For all the tasks conducted in the paper (i.e., route prediction, time prediction, and spatio-temporal graph forecasting), we split the data into 6:2:2 according to the time as the training set, validation set, and test set.

- **Are there any errors, sources of noise, or redundancies in the dataset?** If so, please provide a description.

No.

- **Is the dataset self-contained, or does it link to or otherwise rely on external resources (e.g., websites, tweets, other datasets)?** If it links to or relies on external resources, a) are there guarantees that they will exist, and remain constant, over time; b) are there official archival versions of the complete dataset (i.e., including the external resources as they existed at the time the dataset was created); c) are there any restrictions (e.g., licenses, fees) associated with any of the external resources that might apply to a dataset consumer? Please provide descriptions of all external resources and any restrictions associated with them, as well as links or other access points, as appropriate.

The dataset is entirely self-contained.

- **Does the dataset contain data that might be considered confidential (e.g., data that is protected by legal privilege or by doctor–patient confidentiality, data that includes the content of individuals' nonpublic communications)?** If so, please provide a description.

No.

- **Does the dataset contain data that, if viewed directly, might be offensive, insulting, threatening, or might otherwise cause anxiety?** If so, please describe why.

No.

**Does the dataset identify any subpopulations (e.g., by age, gender)?** If so, please describe how these subpopulations are identified and provide a description of their respective distributions within the dataset.

No.

**Is it possible to identify individuals (i.e., one or more natural persons), either directly or indirectly (i.e., in combination with other data) from the dataset?** If so, please describe how.

Our data has been strictly desensitized and cannot be linked to real individuals.

- **Does the dataset contain data that might be considered sensitive in any way (e.g., data that reveals race or ethnic origins, sexual orientations, religious beliefs, political opinions or union memberships, or locations; financial or health data; biometric or genetic data; forms of government identification, such as social security numbers; criminal history)?** If so, please provide a description.

No.

## 9.3 Collection Process

- **How was the data associated with each instance acquired? Was the data directly observable (e.g., raw text, movie ratings), reported by subjects (e.g., survey responses), or indirectly inferred/derived from other data (e.g., part-of-speech tags, model-based guesses for age or language)?** If the data was reported by subjects or indirectly inferred/derived from other data, was the data validated/verified? If so, please describe how.

The data was observable from the courier's pick-up/delivery data on the (company name blinded) platform.

- **What mechanisms or procedures were used to collect the data (e.g., hardware apparatuses or sensors, manual human curation, software programs, software APIs)?** How were these mechanisms or procedures validated?

The data is collected by the software program in the (company name blinded) platform.

- **If the dataset is a sample from a larger set, what was the sampling strategy (e.g., deterministic, probabilistic with specific sampling probabilities)?**

We pick out several cities and randomly select regions in different cities.

- **Who was involved in the data collection process (e.g., students, crowd workers, contractors) and how were they compensated (e.g., how much were crowdworkers paid)?**

The employees in (company name blinded).

- **Over what timeframe was the data collected?** Does this timeframe match the creation timeframe of the data associated with the instances (e.g., recent crawl of old news articles)? If not, please describe the timeframe in which the data associated with the instances was created.

This data was extracted from the (company name blinded) platform between May and November of a recent year.

- **Did you collect the data from the individuals in question directly, or obtain it via third parties or other sources (e.g., websites)?**

The data was collected from the (company name blinded) platform.

## 9.4 Preprocessing/cleaning/labeling

- **Was any preprocessing/cleaning/labeling of the data done (e.g., discretization or bucketing, tokenization, part-of-speech tagging, SIFT feature extraction, removal of instances, processing of missing values)?** If so, please provide a description. If not, you may skip the remaining questions in this section.

For the privacy issue, no customer-related information (such as the address, name, and ID) is contained in the dataset. And for the couriers, they are our staff, we have got their permission to collect,

and analyze the dataset. Specifically, the following operations are adopted to further protect privacy: 1) a package is not linked to a customer in the dataset to protect the privacy of customers. 2) Instead of using the latitude and longitude, we utilize the coordinate of a package in a two-dimensional space $(x, y)$ to represent a package's location. The relative distance of two locations in real geographical space is preserved in the two-dimensional space. In this way, $(x, y)$ cannot be projected back to a real-world location, thus largely migrating the privacy issue. 3) For couriers, no sensitive information (e.g., gender and age) is included either.

- **Was the "raw" data saved in addition to the preprocessed/cleaned/labeled data (e.g., to support unanticipated future uses)?** If so, please provide a link or other access point to the "raw" data.

No.

## 9.5 USES

- **Has the dataset been used for any tasks already?** If so, please provide a description.

No.

- **Is there a repository that links to any or all papers or systems that use the dataset?** If so, please provide a link or other access point.

Yes.

- **What (other) tasks could the dataset be used for?**

The dataset can be used for route prediction, estimated time of arrival prediction, spatio-temporal graph forcasting, and route optimization. See section 4.4 for more details.

## 9.6 DISTRIBUTION

- **How will the dataset will be distributed** (e.g., tarball on website, API, GitHub)? Does the dataset have a digital object identifier (DOI)?

The dataset will be made available on the internet. There will be a corresponding Hugging Face repository associated with the dataset, and code on how to use the dataset and baseline methods.

- **When will the dataset be distributed?**

The dataset is available to the reviewers and the public along with the submission with a companion Hugging Face repository.

- **Will the dataset be distributed under a copyright or other intellectual property (IP) license, and/or under applicable terms of use (ToU)?** If so, please describe this license and/or ToU, and provide a link or other access point to, or otherwise reproduce, any relevant licensing terms or ToU, as well as any fees associated with these restrictions.

This dataset is licensed under a CC BY-NC 4.0 International License [2]. There is a request to cite the corresponding paper if the dataset is used.

**Have any third parties imposed IP-based or other restrictions on the data associated with the instances?** If so, please describe these restrictions, and provide a link or other access point to, or otherwise reproduce, any relevant licensing terms, as well as any fees associated with these restrictions.

No.

**Do any export controls or other regulatory restrictions apply to the dataset or to individual instances?** If so, please describe these restrictions, and provide a link or other access point to, or otherwise reproduce, any supporting documentation.

---

[2]https://creativecommons.org/licenses/by-nc/4.0/

No.

## 9.7 MAINTENANCE

- **Who will be supporting/hosting/maintaining the dataset?**

The employee in (company name blinded) will host the dataset on Hugging Face.

- **How can the owner/curator/manager of the dataset be contacted (e.g., email address)?**

The authors can be contacted via their emails mentioned in the paper.

- **Is there an erratum?** If so, please provide a link or other access point.

Not to our best knowledge.

- **Will the dataset be updated (e.g., to correct labeling errors, add new instances, delete instances)?** If so, please describe how often, by whom, and how updates will be communicated to dataset consumers (e.g., mailing list, GitHub)?

The corresponding Hugging Face page will be updated regularly.

- **Will older versions of the dataset continue to be supported/hosted/maintained?** If so, please describe how. If not, please describe how its obsolescence will be communicated to dataset consumers.

The old versions of the dataset will not be maintained. If we update the version of the dataset, we will put the specific details of the dataset update on the relevant Hugging Face.

- **If others want to extend/augment/build on/contribute to the dataset, is there a mechanism for them to do so?** If so, please provide a description. Will these contributions be validated/verified? If so, please describe how. If not, why not? Is there a process for communicating/distributing these contributions to dataset consumers? If so, please provide a description.

If others want to extend/augment/build on/contribute to the dataset, please contact the original authors about incorporating fixes/extensions.

