# OpenReview forum: "LaDe: The First Comprehensive Last-mile Express Dataset from Industry"
_ICLR.cc/2024/Conference — Submitted to ICLR 2024_

### Official Review · Reviewer_7pge · 2023-10-31

**Soundness:** 3 good
**Presentation:** 3 good
**Contribution:** 3 good
**Rating:** 8
**Confidence:** 4

**Summary:**

This paper introduces “LaDe”, a publicly available real last-mile delivery dataset collected from five cities in China in time span of 6 months. Both pickup and delivery instances are provided separately with other comprehensive information such as which courier picked up or delivered the package, the location of the package, and the corresponding time. This dataset addresses the limitations of previous last-mile delivery datasets and has the potential to facilitate advancements in logistics and supply chain research. The authors compare this dataset on three tasks and also mention other tasks where this dataset could be used.

**Strengths:**

There was a lack of widely acceptable and publicly available real last-mile delivery dataset. Earlier datasets such as Amazon last mile delivery dataset was at small scale. “LaDe” seems to be the biggest real last-mile delivery dataset till now, with separate pickup and delivery instances.
1. This dataset may help many researchers in the domain of logistics, supply chain management, spatio-temporal data mining, etc.
2. The paper is well-written explaining the delivery process, defining the dataset, and prospective use of the dataset for various tasks.
3. The paper conducted experiments on three tasks including route prediction, estimated time of arrival prediction, and spatio-temporal graph forecasting.
4. The paper highlights the limitations of existing datasetsok and compares it with their proposed dataset, “LaDe”.

**Weaknesses:**

1. The limited geographic coverage of this dataset in five Chinese cities raises concerns about its broader relevance and transferability to different geographic and cultural contexts.
2. Another limitation could be missing timestamp and location of all the trajectories covered by couriers, currently, only pickup and drop locations and corresponding time are recorded.

**Questions:**

1. The related work survey for the benchmarked tasks could be made more comprehensive as well. In route prediction, several works exist. Some missing citations would be [1-3].
* [1] Jain, Jayant, Vrittika, Bagadia, Sahil, Manchanda, Sayan, Ranu. "NeuroMLR: Robust & Reliable Route Recommendation on Road Networks". NeurIPS 2021.
* [2] Hao Wu, Ziyang Chen, Weiwei Sun, Baihua Zheng, and Wei Wang. Modeling trajectories with recurrent neural networks. In IJCAI, pages 3083– 3090, 2017.
* [3] Xiucheng Li, Gao Cong, and Yun Cheng. Spatial transition learning on road networks with deep probabilistic models. In ICDE, pages 349–360, 2020.

2. Can the font of axis labels on images/graphs in figures 3 and 4 be enlarged?

3. While the dataset's findings may be valuable for last-mile delivery research, it's crucial for the authors to critically evaluate its broader utility and acknowledge any limitations.The authors can try to include several points:
* Geographic Diversity: Last-mile delivery challenges can vary significantly across regions, influenced by factors like urban layout, road infrastructure, population density, and geographical features. It's essential to explore how well the dataset's findings can be extrapolated to cities with different characteristics, such as those in other countries or even within China.
* Cultural Factors: The cultural norms and consumer behavior in China might differ from those in other countries. How do these cultural factors affect the dataset's applicability in regions with distinct consumer preferences or expectations? This could include considerations related to delivery timeframes, delivery modes, and consumer attitudes toward delivery services.
* Environmental Factors: Environmental conditions, including weather, air quality, and traffic congestion, play a significant role in last-mile delivery operations. The authors should analyze whether the dataset's findings are sensitive to these environmental factors and whether the methods proposed can adapt to varying conditions in different regions.
* Regulatory and Infrastructure Variances: Different countries and regions may have unique regulatory frameworks and transportation infrastructure. How might these variations impact the implementation and effectiveness of the last-mile delivery strategies derived from the dataset?
* Socio-Economic Disparities: Socio-economic disparities can affect last-mile delivery in various ways. It's important to assess whether the dataset captures these disparities and how they may differ in other regions. Are the strategies proposed in the study suitable for addressing disparities in different socio-economic contexts?

Including the above points will enhance the practical relevance and impact of the research for a wider audience and encourage further discussion on adapting and extending the findings to different international settings.

---

> ### Author Response · Authors · 2023-11-20
>
> Thanks for your constructive comments!
>
> Q1-Q2: Thanks for the comment. We will add the related works and revise the figure in the revision.
>
> Q3: Thanks for the valuable suggestions. We will improve the dataset by considering the given points.

---

### Official Review · Reviewer_f5QW · 2023-10-31

**Soundness:** 3 good
**Presentation:** 3 good
**Contribution:** 3 good
**Rating:** 5
**Confidence:** 4

**Summary:**

This paper a large-scale real world last mile express dataset, which can be used for research in logistics, supply chain management, and spatio-temporal data mining. This dataset includes 10,677k packages of 21k couriers over 6 months of real world operation. Moreover, this dataset also include comprehensive information of the packages, and data from various scenarios. The authors also perform experiments to verify the performance of three different tasks on the proposed datasets.

**Strengths:**

1. The authors introduce a large-scale last-mile express dataset from industry to support the research.

2. The proposed dataset includes comprehensive information and data from different scenarios. It can support different research topics, including logistics, supply chain management, and spatio-temporal data mining.

3. The authors have performed extensive experiments to demonstrate the effectiveness of the proposed dataset for different research topics.

4. This paper is clearly written and easy to follow.

**Weaknesses:**

1. Although the proposed dataset can support the research in several research topics, it can only support a narrow research area in AI. Thus, its impacts may be limited.

2. The proposed dataset is more suitable for data mining tasks. It may not be very relevant to this conference, which focuses on more general deep learning methods. It should be better to release this dataset in a data mining conference/journal.

**Questions:**

1. Besides the logistics, supply chain management, and spatio-temporal data mining, are there any other deep learning related research areas that can use this dataset for experimental evaluation?

2. This dataset is collected in several cities in China. Will the models developed on this dataset be suitable to the applications in other countries?

---

> ### Author Response · Authors · 2023-11-20
>
> Thanks for your constructive comments, we address your questioins in the following points:
>
> Q1:Thanks for the comments. In addition to primary tasks, the dataset can provide substantial support for a wide range of other tasks in different research fields. Firstly, LaDe can be used for representation learning, which involves many topics that can be broadly classified by the representation object: i) POI (Point of Interest) representation learning , i.e., to learn the representation of the pickup/delivery location. ii) trajectory representation learning, i.e., to learn the representation of courier trajectory; ii) AOI representation learning. Secondly, LaDe can be utilized to verify algorithms for optimization problems, such as the vehicle route problem, delivery scheduling problem. Moreover, benefiting from its large data volume and detailed information, LaDe shows great potential to support the development of foundation models in geo-related domains.
>
>
> Q2: Thanks for the comments. We acknowledge that LaDe may be limited to one country. However,  it is quite difficult for a single company to collect data from different countries. To address this challenge, we record data from different cities with different characteristics so that models developed in this dataset can generalize well in different scenarios (such as in different countries). Many famous datasets only contain data from one country, such as the NYC taxi data (https://github.com/toddwschneider/nyc-taxi-data) and data mentioned in related work.  Our work contributes the first pick-up and delivery dataset from the industry,  which should not be neglected mainly because it is from one country.

---

### Official Review · Reviewer_CAYh · 2023-11-04

**Soundness:** 2 fair
**Presentation:** 3 good
**Contribution:** 2 fair
**Rating:** 3
**Confidence:** 5

**Summary:**

This paper releases a dataset named LaDe, collected from a logistics company. The authors describe the information in the dataset, conduct some analyses, and test performance in three applications.

Overall, there are some issues with this paper.

1. The paper is not in the scope of ICLR. ICLR is a top-tier venue for sharing the development of machine learning and its applications. The released dataset has a limited contribution to the machine learning community. Although the authors have discussed three applications, they are not the main topic of ICLR.

2. The dataset is collected based on the real-world service on the logistics platform, and thus it is not general. The authors have not considered an important question: what about other logistics platforms with different express services? Will the dataset be useful?

3. I suggest the authors propose novel machine learning methodologies for machine learning tasks, including classification, prediction, etc., for this dataset.

**Strengths:**

1. The paper is well written.
2. The relased link provides detailed descriptions for the dataset.

**Weaknesses:**

1. The paper is not in the scope of ICLR. ICLR is a top-tier venue for sharing the development of machine learning and its applications. The released dataset has a limited contribution to the machine learning community. Although the authors have discussed three applications, they are not the main topic of ICLR.

2. The dataset is collected based on the real-world service on the logistics platform, and thus it is not general. The authors have not considered an important question: what about other logistics platforms with different express services? Will the dataset be useful?

3. I suggest the authors propose novel machine learning methodologies for machine learning tasks, including classification, prediction, etc., for this dataset.

**Questions:**

See comments above.

**Details Of Ethics Concerns:**

The collected data may reveal private information for invovled users.

---

### Official Review · Reviewer_daoa · 2023-11-07

**Soundness:** 2 fair
**Presentation:** 3 good
**Contribution:** 2 fair
**Rating:** 5
**Confidence:** 4

**Summary:**

The paper introduces "LaDe," which is claimed to be the first publicly available last-mile delivery dataset, encompassing a substantial dataset of 10,677,000 packages delivered by 21,000 couriers over a six-month duration. This dataset holds the potential to establish a benchmark for the evaluation of route prediction, estimated time of arrival prediction, and spatio-temporal graph forecasting tasks.

**Strengths:**

S1. The provision of an open-access, real-world dataset in this article has a favorable and constructive influence on research within the domain of Pickup & Delivery.

S2. Following the release of the dataset associated with this article, it can be employed to corroborate and verify findings from research papers that had not previously made their datasets openly available.

S3. The dataset introduced in this article incorporates measures to protect crucial user privacy information.

**Weaknesses:**

W1. The data fields of the dataset proposed in the article appear relatively straightforward and, in essence, do not significantly distinguish themselves from the data fields found in other existing Pick-up & Delivery datasets.

W2. The dataset outlined in this research paper reveals certain limitations with respect to its incorporation of road network structure, a pivotal aspect for applications pertaining to pick-up and delivery services. Predominantly relying on GPS locations(x-y coordinates), while undoubtedly valuable, might not comprehensively address the intricacies inherent in real-world logistics operations. The absence of road network information in the dataset does raise valid concerns about its suitability for applications necessitating an in-depth understanding of spatial constraints and the optimization of routing, particularly in the last-mile express domain. To augment the dataset's applicability in such scenarios, it may prove advantageous to contemplate the inclusion of coarse-grained road network data. While acknowledging the privacy concerns associated with road network information, providing a more generalized form of road network data could still facilitate researchers in achieving a more comprehensive and accurate representation of the real-world scenarios under examination.

W3. The dataset should acknowledge the significant impact of rush hours and traffic congestion on logistics. It is important to recognize that the road network structure undergoes dynamic and temporal changes over the course of several months. Therefore, it becomes paramount to incorporate a dynamic road network structure within the dataset that can adapt to and depict the wide-ranging traffic conditions and road configurations at different times. This dynamic approach not only bolsters the dataset's precision and relevance but also furnishes researchers with a more authentic representation of the continually evolving real-world logistics and last-mile delivery scenarios.

W4. It is worth noting that several datasets, including prominent examples like the Didi Chuxing GAIA dataset and the New York City dataset, have been extensively employed in addressing pick-up and delivery problems. These datasets have formed the foundation of numerous benchmarks and research benchmarks, rendering them well-established within the field. Consequently, it is essential to delineate the distinctive advantages that the dataset introduced in this paper brings to the table in comparison to these existing datasets and benchmarks, particularly in the context of last-mile delivery research. Clarifying the unique attributes and contributions of this new dataset is crucial in highlighting its relevance and significance within the academic and research community. To accentuate the contributions of the article, it is essential to delve deeper into a comprehensive comparison that delineates the differences between the dataset tailored for Last Mile Delivery scenarios and datasets associated with broader Pick-up & Delivery contexts, including well-known examples like the DIDI and NYC taxi datasets.

W5. Throughout the article, the authors seek to benchmark several aspects of applications, each of which boasts a significant body of existing works. Yet, the chosen methods in the article appear to fall short of encompassing the full spectrum of relevant works and categories within these benchmarked areas. In light of this, it is imperative that the authors offer more extensive elucidation regarding their decision to select these specific methods as baselines when there are numerous alternatives at their disposal. Providing a comprehensive explanation will not only enhance the transparency and academic rigor of their research but also enable the reader to grasp the underlying reasoning behind these choices.

**Questions:**

Beyond the aforementioned weak points, the additional question is as follows.

Q1. Does the proposed benchmark potentially unveil latent insights or yield noteworthy contributions to the field? Moreover, what benefits can be discerned from conducting experiments with this dataset in contrast to other pre-existing datasets?

---

> ### Author Response · Authors · 2023-11-20
>
> Thanks for your constructive comments, we address your concerns in the following points:
>
> W1: Thanks for the comments. We'd like to argue that our compared with previous works,  we are the first publicly available last-mile dataset containing both package pick-up and delivery data.
>
> W2: Thanks for the valuable suggestion. We will add the road network to the dataset in the revision.
>
> W3: Thanks for the suggestions. There are several differences in our data compared with DIDI and NYC taxi data: i) different service patterns. In datasets like DIDI, drivers usually only pick up or deliver one customer at a time, while in our dataset, a courier can have several packages to pick up or deliver at the same time, which is more challenging considering the spatial-temporal constraints. ii)   more comprehensive information. NYC taxi data only records pick-up and drop-off time&location, while we provide detailed trajectory during the courier's service process. Besides, compared with  NYC and DIDI, we also provide the worker id (i.e., courier id) and AOI related information, so that more tasks can be conducted related to a worker or the geographical space. iii) More diverse. DIDI contains data from two cities, and NYC only provides data from one city. In contrast, we provide data from 5 different cities, which are more diverse with different spatial-temporal patterns.
>
> W5. Thanks for the comments. For the route and time prediction task, we argue that we have implemented all related baselines. And for the STG prediction task, there are indeed numerous models proposed in recent years, and it is hard to include all of them considering the limited page of the paper. Therefore, we implemented several basic models (i.e., HA, DCRNN, STGCN), the most popular models (i.e., ASTGCN, MTGNN, AGCRN), as well as the most recent SOTA models (i.e., STGNCDE, GMSDR) as the baselines.

---

### Meta-Review · Area_Chair_xLbm · 2023-12-04

**Metareview:**

In this paper, the author introduces a dataset named LaDe which is claimed to be the first publicly available last-mile delivery dataset and has three unique characteristics, large-scale, comprehensive information, and diversity.

We got four reviews with ratings of 5, 3, 5, and 8 with confidence of 4, 5, 4, and 4 respectively. Reviewers (daoa, 7pge) acknowledge the constructive influence of this article on relevant research.  However, reviewers (daoa, CAYh, f5QW) suggest that the contribution and the possible impact of this article are somewhat limited. The final decision reject.

**Justification For Why Not Higher Score:**

The contribution and the possible impact of this article are somewhat limited

**Justification For Why Not Lower Score:**

N/A

---

### Decision · Program_Chairs · 2024-01-16

Reject